# Development of Novel High and Low Emulsifier Diets Based upon Emulsifier Distribution in the Australian Food Supply for Intervention Studies in Crohn’s Disease

**DOI:** 10.3390/nu16121922

**Published:** 2024-06-18

**Authors:** Jessica A. Fitzpatrick, Peter R. Gibson, Kirstin M. Taylor, Emma P. Halmos

**Affiliations:** 1Department of Gastroenterology, School of Translational Medicine, Monash University, Melbourne 3004, Australia; peter.gibson@monash.edu (P.R.G.); emma.halmos@monash.edu (E.P.H.); 2Department of Gastroenterology, Alfred Health, Melbourne 3004, Australia

**Keywords:** emulsifiers, food additives, Crohn’s disease, diet, diet therapy

## Abstract

Background: The aims of this study were to develop and evaluate a high/low-emulsifier diet and compare emulsifier content with preclinical studies that have associated Crohn’s disease with emulsifiers. Methods: Supermarkets were audited with a seven-day high- (HED) and low-emulsifier diet (LED) meal plan developed. The emulsifier content of food was sought from food manufacturers, compared to acceptable daily intake (ADI), and doses were provided in trials. Nutritional composition analysis was completed. Healthy adults ate these diets for seven days in a randomized single-blinded cross-over feeding study to assess palatability, tolerability, satiety, food variety, dietary adherence, blinding and the ease of following the meal plan via visual analogue scale. Results: A database of 1680 foods was created. There was no difference in nutritional content between the HED and LED, except HED had a higher ultra-processed food content (*p* < 0.001). The HED contained 41 emulsifiers, with 53% of the products able to be quantified for emulsifiers (2.8 g/d), which did not exceed the ADI, was similar to that in observational studies, and was exceeded by doses used in experimental studies. In ten participants, diets were rated similarly in palatability—HED mean 62 (5% CI 37–86) mm vs. LED 68 (54–82) mm—in tolerability—HED 41 (20–61) mm vs. LED 55 (37–73) mm—and in satiety HED 57 (32–81) mm vs. LED 49 (24–73) mm. The combined diets were easy to follow (82 (67–97) mm) with good variety (65 (47–81)) and excellent adherence. Conclusion: Nutritionally well-matched HED and LED were successfully developed, palatable and well tolerated.

## 1. Introduction

The development of Crohn’s disease has been associated with high consumption of ultra-processed foods that contain, amongst other non-nutritive ingredients, dietary emulsifiers and thickeners [1]. Some of these emulsifiers and thickeners have gained attention from pre-clinical models as potential drivers of intestinal inflammation yet little is known about their effect in humans [2]. The three thickeners and two emulsifiers that have been evaluated in pre-clinical models are carboxymethylcellulose (CMC), maltodextrin, carrageenan, polysorbate 80 and polysorbate 60 [2,3]. Studies in these models have demonstrated a shift to inflammatory microbiota with increased flagella, reduced mucus thickness allowing bacterial encroachment to the gastrointestinal epithelium and increased intestinal permeability, allowing increased bacterial translocation and development of colitis [2].

These exploratory models are noteworthy but are difficult to extrapolate to dietary intake in humans with Crohn’s disease for several reasons. First, the emulsifier is delivered outside the context of food, such as through the drinking water of mice or via supplementation [3,4]. Second, the doses of these food additives per kilogram of body weight are very high compared with those approved for use in humans. Third, pre-clinical models have only assessed short-term exposure to one emulsifier at a time. Fourth, the categories of emulsifiers fed to the animals can be a different grade to those added to food, as in the case of carrageenan [2]. Finally, in relation to food, they are often added to ultra-processed food that may have other deleterious compounds driving intestinal changes. Paradoxically, our best dietary treatment for inducing endoscopic remission in Crohn’s disease, exclusive enteral nutrition, contains emulsifiers and thickeners, including the ones heavily studied in pre-clinical models [5].

Despite the literature implicating ‘emulsifiers’ in Crohn’s disease, researchers are actually referring to a class of both emulsifiers and thickeners that are naturally occurring (e.g., egg yolk or milk casein and whey) or additives that are added to packaged food to create stable lipid and water emulsions, improve the texture, palatability, appearance or mouthfeel of a product, act as a weighting agent or enhance the product’s thickness in the absence of dietary fat [6].

Currently, there are two dietary strategies used to induce remission in Crohn’s disease; however, they are both highly restrictive, and without the guidance of a specialised IBD dietitian, they can be difficult for patients to adhere to [7,8]. These therapies are exclusive enteral nutrition and the Crohn’s disease exclusion diet [9,10]. They involve drinking high volumes of oral nutrition supplements and following either a strict list of foods in the case of the Crohn’s disease exclusion diet or exclusion of food all together in the case of exclusive enteral nutrition. Whilst these dietary therapies are efficacious, they are not sustainable, and their mechanisms are poorly understood [11]. Therefore, novel whole-food dietary strategies are needed that are sustainable, well tolerated, and understood in terms of their mechanisms.

To date, no whole diet strategy has been described to evaluate the effect this class of food additives has on human gut physiology. Well-described and evaluated dietary trials using high- or low-emulsifier diets are needed to determine if these pre-clinical models translate to impaired intestinal barrier function and intestinal inflammation in humans. Therefore, the aims of the study were (a) to develop novel high- and low-emulsifier diets based on commercially available ingredients and foods, reflective of the Australian dietary guidelines and matched on all dietary components except emulsifiers and thickeners; (b) to evaluate their suitability for future interventional studies, such as in patients with Crohn’s disease, by defining, in healthy subjects, tolerance, blinding and adherence to the diets; and (c) to interrogate the emulsifier and thickener content of an emulsifier-rich Australian diet to enable comparison with experimental dietary intake, reported daily exposure from observational studies and acceptable daily intakes (ADI) [12].

## 2. Materials and Methods

### 2.1. Diet Design and Preparation

The aims of dietary design were to create diets high and low in emulsifiers based upon readily available Australian foods that are widely consumed while avoiding the introduction of confounders such as nutritional inadequacy or other major compositional differences. Hence, the criteria set for the design of the diets included that they met Australian dietary guidelines [13]; were matched in all macronutrients, micronutrients and fibre; were based on ingredients and packaged food products readily available in major supermarkets in Australia, and differed only in the presence or absence of dietary emulsifiers/thickeners. For packaged food products, emulsifiers were deemed to be present based on the ingredients listed on back-of-label packaging using CODEX e numbers [6]. Thickeners and emulsifiers were identified as egg products 322, 331, 327, 400–499, gelatine; whey; shortening; hydrolyzed vegetable protein; hydrolyzed milk protein; psyllium; dextrin and modified starch, including 1404, 1412, 1422, 1442 and 1520. Whole egg was also excluded in the low-emulsifier diet.

In order to develop the diets, audits of seven different Australian supermarkets were undertaken in September 2018. Packaged foods were classified into confectionery, biscuits, spreads, condiments, ice cream, soft drinks, bread, cakes/pastry and dairy, creating a database of readily available emulsifier-free and emulsifier-containing foods using their ingredients list. Food containing additives of interest were identified, including emulsifiers used in pre-clinical studies (CMC, polysorbate 80, polysorbate 60, carrageenan), in addition to additives of other classifications, such as preservatives and acidity regulators. This database was then used to develop a seven-day meal plan of two diets—a high-emulsifier diet (HED) using products available in supermarkets only, in which every ingredient possible and meal contained emulsifiers from the previously described additives, and a matched low-emulsifier diet (LED) that avoided all of the described emulsifier and thickener additives but followed the same meal plan. Where possible, products containing the greatest number of emulsifiers/thickeners were chosen for the HED. Soft drinks were excluded from both diets as often weighting agents are used as compound ingredients that are not required to be identified on the ingredients list [6]. Seventeen recipes were created in consultation with a research chef, who supervised the preparation of meals. The recipes were prepared by the research chef in accordance with food safety guidelines, portioned, vacuum-sealed and frozen. All dry consumables were taken out of the packet, portioned and vacuum sealed and relabelled with study labels (Figure 1) to facilitate blinding.

### 2.2. Analysis of the Nutritional Composition of the Diets

The macro- and micronutrient content of the diets was analysed with FoodWorks (Xyris software version 10), FODMAP content was determined using the FODMAP online calculator (https://monashfodmapcalculator.com.au (accessed on 22 January 2020)) and the proportion of ultra-processed foods estimated using the NOVA system [14]. In order to quantify the content of emulsifiers and thickeners in the HED, data were sought from food manufacturers. Where provided, the percentage of emulsifier/thickener content of packaged foods was multiplied by the quantity of food prescribed in the seven-day meal plan (g), then divided by seven to calculate the daily provision of each emulsifier/thickener and converted to mg/d. The daily total content of emulsifiers and thickeners was calculated by adding the daily provision of each emulsifier/thickener together. These data were compared with the ADI, defined as estimate of the amount of a substance in food or drinking water that can be consumed daily over a lifetime without presenting an appreciable risk to health, set by the European Food Safety Authority [15].

The HED content of food additives of interest (CMC, polysorbate 80 and carrageenan) was compared to the amount of these additives provided in preclinical and clinical trials. To determine the dose provided in pre-clinical trials in g/d, the percentage of these additives supplemented to the drinking water in the murine model was used, with assumptions regarding mouse weight and fluid consumed, where this was not detailed in the study. For clinical trials, the doses provided in the supplements were used. Daily exposure to these emulsifiers of interest on the HED was compared to observational data where this has been reported [16]. Where no data were available for these food additives of interest in the HED (i.e., not provided by the food manufacturer), estimates of ‘mean reported use levels (mg/kg)’ were used from the European Food Safety Authority using the most similar food category. This mg/kg of food additive was adjusted for the grams of packaged food product provided in the seven-day meal plan and then divided by seven to calculate the daily provision of each emulsifier/thickener (mg/d).

### 2.3. Evaluation of High- and Low-Emulsifier Diets in Pilot Study

#### 2.3.1. Participants

Healthy volunteers were sought between January–March 2020 by posting study recruitment flyers on noticeboards at Alfred Health. Electronic versions were posted on Monash University FODMAP social media pages, study website and study coordinators’ social media accounts. Inclusion criteria for healthy participants were aged between 18 and 60 years with no major medical conditions. Exclusion criteria were restrictive diets (including vegetarian/vegan), antibiotics, probiotics or prebiotics within the last two weeks, pregnant/breastfeeding, extremes of body mass index (<18.5 kg/m^2^ or >35 kg/m^2^), taking medical therapy known to affect gastrointestinal function or gut microbiota, significant mental health illness and inability to give informed consent. Informed consent was obtained prior to participants commencing the study.

#### 2.3.2. Study Protocol

In a single-blinded, cross-over, randomized control study, 10 healthy subjects were provided with one week each of HED or LED with ≥3-week washout between diets. Randomization was allocated using a sequential randomization sequence generated via http://randomization.com/ (accessed on 22 January 2020) (sequence 27370, created 22 January 2020). Participants were blinded to their dietary allocation. The study protocol was approved by Alfred Health Ethics Committee.

At the end of seven days, participants completed an electronic Redcaps survey (https://redcap.alfredhealth.org.au/redcap/surveys/?s=X3WJCTYDFD (accessed on 22 January 2020)) using a 100 mm visual analogue scale (0 = very poor to 100 = very good) to assess both diets for palatability (i.e., taste, texture, smell, appearance), tolerability (i.e., how the diet made participants feel) and satiety (i.e., how satisfied participants were with the amount of food they received). On a similar scale, participants were asked to rate the food variety in the overall meal plans, the ease of following the meal plan and willingness to participate again using these diets. To assess blinding, participants were asked which diet they thought would be an interventional diet to treat Crohn’s Disease as well as the qualitative reasons behind their choice with a free text box for their response. They were asked to provide any feedback on how the meal plans could be improved with a free text box for their reply.

Adherence to the diet was assessed by a seven-day food diary. Adherence was defined as consuming >95% of food provided, to the exclusion of all other non-permitted foods.

### 2.4. Statistical Analysis

Descriptive statistics were presented as mean and 95% confidence interval or median and range depending on distribution. Differences between paired data were assessed by paired *t* test. Student *t* tests were used to compare daily nutrient provision between diets, and those with a *p*-value ≤ 0.003 were considered statistically significant after Bonferroni correction for multiple comparisons. As this was a pilot trial with no comparison data, a power calculation was not made. Ten participants were considered to be sufficient on the basis of previous pilot evaluations of a diet [17]. Differences in pilot data were considered statistically significant at a *p*-value ≤ 0.05. Prism software (version 9.5.1 (528); GraphPad Software, San Diego, CA, USA) was used for analysis.

## 3. Results

### 3.1. Design of Novel Diets

The supermarket database that contains 1680 individual food items was developed after auditing seven different supermarkets in Australia, and those containing the aforementioned pre-clinical emulsifiers are presented in Table 1. Carrageenan was the most prevalent food additive of interest, being present in 8.2% of items in the database with a high presence in 42.5% of ice cream products, followed by other dairy products (12.6%) and cakes/pastries (8.7%). CMC was the next most prevalent, present in 3.6% of total products, the majority being bread (12.2%). Polysorbate 60 was in 1.4% of products mostly found in cakes and pastries, whereas polysorbate 80 was found in 0.2% (*n* = 3) products. Approximately 25% of the food in the database did not contain any emulsifier or thickener. The seven-day meal plans were designed for the HED and LED using knowledge from this database and using food available at local supermarkets (Table 2).

### 3.2. Evaluation of HED and LED Diets

Ten participants were screened and randomised, and they completed both diets with no drops and were all included in the analysis (Appendix A). They had a median age of 34 (range 12) years and seven males. There were no statistically significant differences in palatability—HED mean 62 (95% CI 37–86) mm vs. LED 68 (54–82) mm—in tolerability—HED 41 (20–61) mm vs. LED 55 (37–73) mm—in satiety HED 57 (32–81) mm vs. LED 49 (24–73) mm (Figure 2). The combined diets were rated easy to follow (82 (67–97) mm) with good variety (65 (47–81)) and excellent adherence, with only one participant eating a non-permitted food on the LED. All participants reported that they were happy to participate in a trial again using the same food. With regards to blinding, all participants accurately chose the LED as the diet they thought would be an interventional diet to treat Crohn’s disease, citing it as ‘less processed’ and more ‘homemade’. Qualitative feedback on how the diets could be improved included increasing the amount/portions of food provided at breakfast and lunch; reconsidering the amount of bread, chickpeas and lentils provided as this caused bloating in some (though not reported as an adverse effect) and reconsidering some of the products in the HED as they were considered to be too sweet for some.

### 3.3. Nutritional Composition of the Diets

The composition of the HED and LED is shown in detail in Table 3. There were no differences in food groups, macronutrients and FODMAP content, with the exception of ultra-processed food (HED: 70% energy/d vs. LED: 29% energy/d, *p* < 0.0001). The total sugar intake tended to be greater in the HED (99 g/d) compared with that in the LED (77 g/d; *p* = 0.03), but this did not reach statistical significance when correction from multiple comparisons was applied (Table 3). The percentage of energy from sugar was not significantly different between the two diets (21% HED and 17% LED).

The HED contained a total of 41 different emulsifiers and thickeners with 28 different emulsifiers, 8 different thickeners and five additives with dual emulsifying and thickening capabilities (Figure 3). Meat-based frozen meals had the most food additives, containing 22 different emulsifiers and thickeners, followed by sweet snacks, dairy spreads and sauces, each containing 10. The most common additives used were xanthan gum present in 37% of products, followed by mono- and diglycerides (27%) and guar gum (22%). Emulsifiers and thickeners of interest were found in a small number of products: carrageenan (407/407a) in 10% (4/41) products, CMC, polysorbate 80 and polysorbate 60 in 2% (1/41) of products each. No emulsifiers or thickeners were present in the LED. Other food additives are described in Table 4.

Quantifying the emulsifier content was achieved by obtaining detailed composition from the manufacturers for 53% of products, which contained 29 different emulsifiers and thickeners (Table 5). None exceeded the ADI that had been set by the European Food Safety Authority, with the daily intake of emulsifiers and thickeners being estimated at 2.8 g/d (Table 5).

### 3.4. Comparison of Emulsifier and Thickener Intake with Those in Pre-Clinical Studies

The intake with the designed HED was compared in terms of frequency with food records in a study examining dietary patterns in children with Crohn’s disease. The mean daily number of exposures with the HED compared with those reported in the Crohn’s disease cohort for carrageenan was 0.85 *vs.* 0.58; for polysorbate 80, it was 0.14 vs. 0.07 and for CMC, it was 0.14 vs. 0.05 [16].

For the unknown proportion of emulsifiers and thickeners of the HED, an estimate was based on the worst-case scenario using the ‘mean reported use levels’ provided by the industry to the European Food Safety Authority (Table 6). HED estimated total intake (mg/d) of carrageenan, polysorbate 80 and carboxymethyl cellulose was higher compared to the median intake of these additives from three-day photographic food records diaries from healthy controls in the ENGIMA study but were still at very low levels [18]. The HED’s estimated intake of these additives compared to their ADI was 33-fold lower and 72,916,666-fold lower for carrageenan and polysorbate 80, respectively (CMC has no ADI) (Table 6).

The HED’s estimated total intake was compared to experimental studies, shown in Table 6, with the experimental murine and human doses exceeding the HED’s estimated intake for all additives by 1.2 to 2.9 × 10^6^-fold.

## 4. Discussion

Detrimental effects of emulsifiers and thickeners on the gastrointestinal tract and gut microbiota have been reported in animal models and in vitro, but their relevance to the human consumption of such additives in the context of food requires evaluation. The dietary design of the diets was successful as they were based upon readily available foods, met dietary guidelines, were well matched in composition except for the targeted food additives and were well tolerated and acceptable in a small cohort of healthy adults. We identified poor blinding between the diets, requiring modifications before application in clinical trials. The semi-quantification of the emulsifier content of the HED indicated that their total and specific intake was less than the intake or exposure in previous experimental studies, often by huge amounts, questioning the relevance of such studies to actual exposure in the Australian population.

Since one aim of designing the diet was to apply them in interventional studies, most dietary confounders were minimized, but the HED tended to have greater total sugar content and, not surprisingly, a higher percentage of energy from ultra-processed food. This further demonstrates the synonymous nature emulsifiers and added sugars have with ultra-processed foods [19,26]. Nutrients that may play defined roles in modulating gastrointestinal physiology, such as fibre, protein, FODMAPs, fat and total caloric provision, were all well matched, typical of an Australian diet [13,27] and reflected Australian dietary guidelines [13]. Therefore, given the complexities when designing a diet for use in clinical trials, these diets were well balanced, and their design would minimise the risk of dietary confounding when applied in future clinical trials [28].

In dietary studies, a major challenge is that blinding of exact food types cannot be performed, and information on food composition is available on the Internet. For these reasons, blinding of the diets was tested and did not perform well with most participants choosing the LED as the future ‘interventional diet’, citing it as less processed. The participants were healthy without specific knowledge of diet and nutrition or of Crohn’s disease, but putative health issues with ultra-processed foods are well publicized. Attempts to disguise this aspect by, for example, removing food from its package, relabelling with study labels and ensuring a minimum three-week washout between diets were only partly successful. Given the participant feedback, adjustments to the novel diets will be required before implementing them in a clinical to improve blinding, whilst not compromising the types and quantity of emulsifiers.

There are multiple issues that challenge the translation of findings of previous studies to real-world diets. First, previous studies have described the effect on intestinal barrier function or inflammation when one emulsifier/thickener is administered at a time in a preclinical setting. Our HED contained 41 different emulsifiers and thickeners in a range of food categories. This is particularly relevant to the preclinical data on emulsifiers that have described the deleterious effects of only a handful of individual emulsifiers and have not examined the potential interaction between additives and other nutrients and their digestibility in the context of food matrixes [29].

Second, it has been touted that carrageenan, CMC, and polysorbate 60 and 80 are ubiquitous in our food supply, yet our HED contained only four food products with carrageenan and CMC, and only one product contained polysorbate 60 and polysorbate 80 aftercare was made to ensure these additives were included in the diet. Our HED was reflective of the foods available in Australian supermarkets, as demonstrated by the supermarket database of 1680 packaged items only having 3 items containing polysorbate 80. Additionally, our HED had a similar daily exposure to a paediatric Crohn’s disease cohort and a healthy control cohort from Australia which showed minimal intake of these additives [16,18]. These highly explored food additives are not an integral part of the Australian food supply and could not be described as ‘widespread’.

Third, the quantity of total emulsifier and individual emulsifier content was considerably lower than what has been described in preclinical models and clinical trials for polysorbate 80, CMC and carrageenan. A challenge in estimating the quantity of emulsifier in the unknown 47% of products on the HED was finding suitable comparisons. Maximal permittable levels from ESFA were not a suitable substitute given some additives (carrageenan, CMC) do not have a maximum permitted level for relevant food categories. So, even using an arbitrary quadrupling of the known 2.8 g/d emulsifier content for the HED to account for the products with unknown emulsifier content estimates a total of 11.2 g/d. This falls short of what has been provided in clinical trials, supplementing with only one food additive, such as a brownie enriched with 15 g CMC provided by Chassaing et al. [22]. Additionally, the murine model doses markedly exceed the HED provision of these additives of interest. It is estimated they provided 2500 mg/kg body weight/day, so for a 70 kg human adult, this would equate to 175 g/day of CMC and polysorbate 80, equivalent to just under 9 tablespoons/d. This demonstrates the pharmaceutical dose of these additives used rather than those relevant to food. Both clinical and pre-clinical doses bear little relationship to what might be consumed in a diet heavily weighted towards such additives.

Fourth, the quantity of total emulsifier and individual emulsifier content in the HED was considerably lower than the ADIs set by the European Food Safety Authority. This may be due to ADIs being developed with a ‘100 times uncertainty factor’ built into their calculations. Even when using the ‘typical reported use levels’ from the European Food Safety Authority, it exceeded our food manufacture data in mg/kg, with the example of carrageenan. The ‘mean reported use levels’ data are solely reliant on the food manufacturers who are willing to share this commercially sensitive information, and as a result, for each food additive within each food category, they often have one product on which to base their data [25].

It remains, however, that the intake of these four additives is likely to be small in Australia when eating food from a supermarket and likely to be only found in particular food categories (such as carrageenan in ice cream), and pre-clinical or supplementary dietary trials cannot be relied upon determine their physiological effect unless their doses are considerably reduced and food additives are provided in the context of food. Indeed, ex vivo studies using the M-SHINE system showed heterogeneous and varied effects on microbiota in flagella expression when the concentrations/doses used were reduced [30].

Our study has some limitations. First, the limited quantitative data of actual food content may have underestimated the amount of emulsifiers and thickeners present in this study. Second, the diet was based on Australian healthy eating guidelines and did not include fast foods and limited discretionary foods, which often are rich in emulsifiers [18]. Third, the inclusion of potentially confounding dietary components cannot be eliminated. Our diets were well matched for all nutrients but differed in the quantities of foods classified as ultra-processed. Given the broad classification of ultra-proceeded foods, this was difficult to avoid as once an emulsifier or thickener is added to a product, it is automatically classified as an ultra-processed food. Finally, despite best efforts, blinding was challenging, and the modifications to the diets have not yet been tested.

## 5. Conclusions

In conclusion, novel and well-matched high- and low-emulsifier diets were successfully developed using foods available to consumers at Australian supermarkets. It is the first to semi-quantify emulsifiers and thickeners in packaged foods and highlights how unrealistic the doses provided in pre-clinical trials have been. The diets had good palatability and were well tolerated by healthy subjects, although minor modifications will be needed in the future before implementation in another clinical trial. The more processed nature of the HED was unable to be disguised. Overall, however, these diets fulfilled most characteristics that make them viable for further clinical trials to determine the effects of variations of the content of emulsifiers and thickeners on gastrointestinal physiology and health with the ultimate aim of determining whether these additives are indeed detrimental (or beneficial) to human health in general or specifically for Crohn’s disease.

## Figures and Tables

**Figure 1 nutrients-16-01922-f001:**
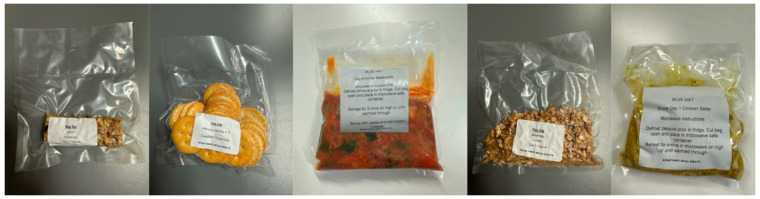
Example of food packaging and labelling for high- and low-emulsifier diets.

**Figure 2 nutrients-16-01922-f002:**
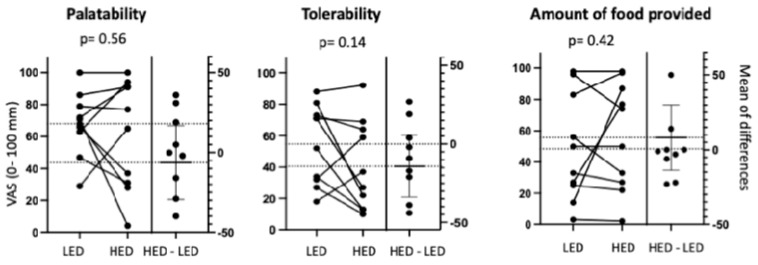
Participant satisfaction in terms of palatability, tolerability and the amount of food provided with the low-emulsifier diet (LED) and high-emulsifier diet (HED) as judged by a visual analogue scale (VAS) where 0 mm represents the lowest score and 100 mm most satisfied.

**Figure 3 nutrients-16-01922-f003:**
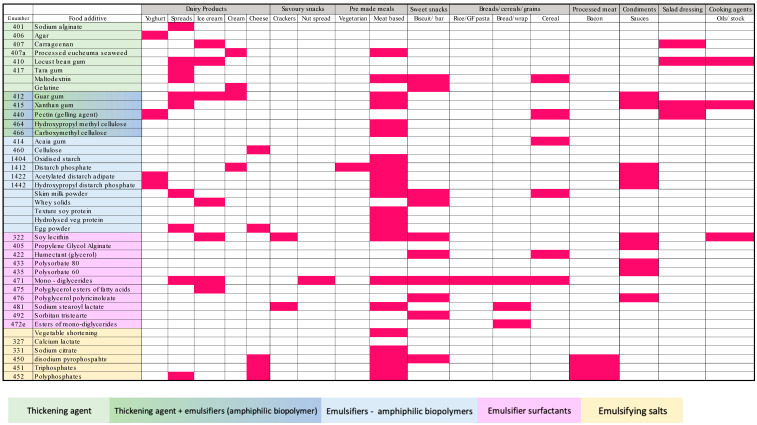
Heat map of emulsifiers and thickeners present in high-emulsifier diet by food categories.

**Table 1 nutrients-16-01922-t001:** Description of the packaged-food database from supermarket audit, categorising food additives of interest and their e-number by food category. n = number of items.

Food Additives of Interest	Polysorbate 80 (433)	Polysorbate 60 (435)	Carboxymethyl Cellulose (466)	Carrageenan (407/407a)	Does Not Contain Any Emulsifiers/ Thickeners	Total Items in Database
Biscuits *n* (%)	0 (0)	0 (0)	0 (0)	0 (0)	6 (5)	121
Confectionery *n* (%)	0 (0)	0 (0)	0 (0)	1 (0.4)	18 (6.8)	265
Spreads *n* (%)	0 (0)	0 (0)	1 (0.5)	1 (0.5)	76 (40)	192
Condiments *n (*%)	2 (1)	3 (1.5)	7 (3.4)	7 (3.4)	65 (31.4)	207
Ice cream *n* (%)	0 (0)	0 (0)	16 (7.5)	91 (42.5)	5 (2.3)	214
Soft drink *n* (%)	0 (0)	0 (0)	0 (0)	0 (0)	106 (95.5)	111
Bread *n* (%)	0 (0)	0 (0)	29 (12.2)	3 (1.3)	75 (31.5)	238
Cakes/pastry *n* (%)	0 (0)	20 (11.6)	6 (3.5)	15 (8.7)	5 (2.9)	173
Dairy *n* (%)	1 (0)	0 (0)	1 (0.6)	20 (12.6)	61 (38.4)	159
Total *n* (%)	3 (0.2)	23 (1.4)	60 (3.6)	138 (8.2)	417 (24.8)	1680

**Table 2 nutrients-16-01922-t002:** Sample seven-day meal plan for high- and low-emulsifier diets.

Day 1	Day 2	Day 3	Day 4	Day 5	Day 6	Day 7
Breakfast						
Cereal ½ cup milk 1 cup milk with drinking chocolate	Instant oats ⅔ cup milk Fruit	Instant oats ⅔ cup milk 1 cup milk with drinking chocolate	Cereal ½ cup milk ¾ cup yoghurt	Cereal ½ cup milk 1 slice bread Peanut butter	Muesli ½ cup milk Fruit	Cereal ½ cup milk 1 slice bread Peanut butter
**Morning Tea**						
Fruit	¾ cup yoghurt	Fruit	Fruit	Fruit	Crackers and Cheese	Fruit
**Lunch**						
2 slices bread with salad filling Muesli bar	Minestrone 2 slices bread	Rice Chickpeas Corn kernels Canned tuna Soy sauce	2 slices bread with salad filling Muesli bar	Chicken & vegetable pie	Mini pizza	Chicken strips Vegetables
**Afternoon Tea**						
Biscuits	Fruit	Crackers and Cheese	Biscuits	Crackers and Dip	Fruit	Crackers and Dip
**Dinner**						
Chicken satay Rice Salad Salad dressing	Lemon & herb fish Stir fry vegetables	Lentil Bolognese Penne pasta Parmesan cheese	Lasagne Salad Salad dressing	Teriyaki salmon Rice Vegetables	Ratatouille Penne pasta Parmesan cheese	Braised lamb shanks Vegetables
**Supper**						
Fruit	Dairy dessert	Fruit	Fruit	¾ cup yoghurt Fruit	Dairy dessert	Fruit

**Table 3 nutrients-16-01922-t003:** Food composition of high- and low-emulsifiers diets expressed per day. Differences were compared using Student *t* test; a *p*-value < 0.003 was considered statistically significant after Bonferroni correction for multiple comparisons. * indicates a statistically significant result.

Nutrient		Diet		*p*-Value for Difference
		High Emulsifier	Low Emulsifier	
Energy (MJ)		8.1	8.3	0.90
Protein	Total (g)	80	92	0.30
	% of energy	16	18	0.26
Fat	Total (g)	75	76	0.97
	% of energy	34	33	0.78
Saturated fat	Total (g)	31	29	0.74
	% of energy	14	13	0.51
Carbohydrate	Total (g)	223	219	0.88
	% of energy	47	43	0.33
Sugars	Total (g)	99	77	0.03
	% of energy	21	17	0.09
Ultra-processed foods	Total (MJ)	5.7	2.3	<0.001 *
	% of energy	70	29	<0.001 *
Fiber (g)		29	33	0.24
Sodium (mg)		2676	2664	0.98
Glucose (g)		21	17	0.24
Lactose (g)		18.8	17.5	0.70
FODMAPs (g)	Total	10.5	9.5	0.71
	Excess fructose (g)	3.1	2.3	0.46
	Sorbitol (g)	1.8	1.7	0.90
	Mannitol (g)	0.9	0.9	0.96
	Fructans (g)	4.0	3.9	0.87
	Galacto-oligosaccharides (g)	0.7	0.7	0.85

**Table 4 nutrients-16-01922-t004:** List of other food additives in the high-emulsifier diet (HED) and low-emulsifier diet (LED).

	LED	HED
Natural flavours		
Natural colours		
Vitamins/Minerals		
Flavour enhancer	635	635
Sweetener	955	955
Acids	300, 330	260, 270,296, 297, 300, 332, 339, 331, 341
Preservatives	202	220, 223, 250, 281
Natural antioxidant	307a	307b, 316
Corn starch		
Oligofructose		
Calcium lactate		327
Potassium Chloride		508
Lysosyme		1105
Pimaricin antifungal		235
Calcium chloride		509
Raising agents		500, 503, 541

LED: low-emulsifier diet, HED: high-emulsifier diet, numbers represent e numbers. 
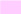
 Absent, 
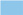
 Present.

**Table 5 nutrients-16-01922-t005:** Quantitative content of emulsifiers and thickeners in the high-emulsifier diet (HED) and comparison to acceptable daily intake (ADI).

Food Additive	e Number	Amount in HED (mg/Day)	ADI (mg/kg/Body Weight)	ADI at 70 kg (mg/d)
**Acetylated distarch adipate (T)**	1422	1614.5	no ADI	
**Whey powder**	No e number	384.1	not listed	
**Hydroxypropyl distarch phosphate (T)**	1442	123.2	no ADI	
**Maltodextrin (T)**	N/A	106.2	not listed	
**Mono- and diglycerides**	471	98.0	no ADI	
**Locust bean gum (T)**	410	74.0	no ADI	
**Distarch phosphate (T)**	1412	71.4	no ADI	
**Agar (T)**	406	70.1	no ADI	
**Soy lecithin**	322	49.1	no ADI	
**Pectin**	440	43.2	no ADI	
**Oxidised starch**	1404	23.6	no ADI	
**Xanthan gum**	415	22.2	no ADI	
**Acacia gum**	414	18.9	no ADI	
**Egg powder**	N/A	17.9	not listed	
**Disodium pyrophosphate**	450	16.2	no ADI	
**Guar gum**	412	13.5	no ADI	
**Humectant (glycerol)**	422	11.3	no ADI	
**Egg**	N/A	5.5	not listed	
**Tara gum (T)**	417	5.2	no ADI	
**Polyglycerol esters of fatty acids**	475	4.7	no ADI	
**Beef Gelatine (T)**	N/A	2.3	not listed	
**Sodium alginate (T)**	401	0.4	no ADI	
**Sorbitan tristearte**	492	20.3	10	700
**Carrageenan (T)**	407	4.4	75	5250
**Sodium stearoyl lactate**	481	4.1	22	1540
**Triphosphates**	451	3.7	40	2800
**Polyphosphates**	452	3.7	40	2800
**Polyglycerol polyricinoleate**	476	1.8	25	1750
**Processed eucheuma seaweed**	407a	1.4	75	5250
**Polysorbate 80**	433	0.0000240	25	1750
Total emulsifier/thickener content (mg/d)		2815		

HED: high-emulsifier diet, ADI: acceptable daily intake as determined by European Food Safety Authority.

**Table 6 nutrients-16-01922-t006:** Comparison of emulsifier and thickener content in high-emulsifier diet to that estimated or provided in published studies.

Emulsifier or Thickener	High-Emulsifier Diet			ADI (mg/d) ^	Median (IQR) Intake in ENIGMA [19] (mg/d)	Intake/Exposure in Experimental Studies (mg/d)	Fold Difference between Experimental vs. HED Total Estimated Intake
	Known Amount (mg/d) *	Estimate of Unknown Amount (mg/d) ^#^	Estimated Total Intake (mg/d)				
Carrageenan (407/407a)	5.78	152 ^#^	157.78	5250	0 (0–960)	200 (human) [20]	1.2
Polysorbate 80 (433)	2.4 × 10^−5^	N/A	2.4 × 10^−5^	1750	0 (0–930)	70 (mice) ‡ [21]	2.9 × 10^6^
Carboxymethyl cellulose (466)	unknown	25.710 ^#^	25.710	N/A	0 (0–800)	70 (mice) ‡ [21] 15,000 (human) [22]	2.7 583

* Data provided by food manufacturer. ^#^ Worst-case scenario estimation using ‘mean reported use levels’ provided by industry to European Food Safety Authority [23,24]. ^ ADI calculated for an average body weight of 70 kg [23,25]. ‡ Assuming a 28 g mouse consumes 7 mL drinking water a day with a 1% infusion of these emulsifiers/thickeners (2500 mg/kg body weight). N/A = non applicable.

## Data Availability

Data described in the manuscript, code book, and analytic code will be made available upon request pending application and approval.

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
