# Peer review of "Development of Novel High and Low Emulsifier Diets Based upon Emulsifier Distribution in the Australian Food Supply for Intervention Studies in Crohn’s Disease"

_nutrients, 2024, doi:10.3390/nu16121922_

Round 1

Reviewer 1 Report

Comments and Suggestions for Authors

In conclusion, novel and well matched high and low emulsifier diets were 375 successfully developed using foods available to consumers at Australian supermarkets

Paper well described but quite restricted to the Australian Supermerket domain

many data reported are innovative and useful to nutrizionist. Tables are informative

Author Response

In conclusion, novel and well matched high and low emulsifier diets were 375 successfully developed using foods available to consumers at Australian supermarkets.

REPLY: Thank you for this feedback.

Paper well described but quite restricted to the Australian Supermerket domain

REPLY: Thank you, yes this study is based on the food supply in Australia. Previous work has been published from the United Kingdom regarding their food supply, which helps to broaden our understanding of the presence of emulsifiers and thickeners in the food supply.

Sandall A, Smith L, Svensen E, Whelan K. Emulsifiers in ultra-processed foods in the UK food supply. Public Health Nutrition. 2023;26(11):2256-2270. doi:10.1017/S1368980023002021

many data reported are innovative and useful to nutrizionist. Tables are informative

REPLY: Thank you, we hope that the readers of Nutrients also find this useful.

Reviewer 2 Report

Comments and Suggestions for Authors

Interesting study. Maybe it will work for more people in the future. I would have a few suggestions.

1. line 17: how many adults were there?

2. line 147: why is >35 kg/m2 an extreme value? It should probably be ≥ 40 kg/m2

3. In the introduction, it is worth writing a few words about why you focused on this disease. Why your diet model may be particularly useful and why other diets for Crohn's disease are, for example, worse. Generally, why your study is innovative. I have the impression that it is not very visible.

4. Why were healthy people included in the study and not people with Crohn's disease? This wasn't what I expected from the title.

5. Incorrect citation in the text, e.g. ".(6)". I think you wrote that everywhere. Please refer to the journal's guidelines for citing articles. I am inserting a fragment from the guidelines "References should be numbered in order of appearance and indicated by a numeral or numerals in square brackets—e.g., [1] or [2,3], or [4–6]. See the end of the document for further details on references."

6. Table 1. "80 (433)" and others in this line. What does it mean. Please give me the legend.

7. Table 2. No grammage (home measurements) provided. Is it intentional?

8. Captions for "Figure 1, Figure 2, Figure 3" are as a screenshot and in the text. This means that the content is repeated. Please remove figure captions from screenshots.
